# COUNTERFACTUAL REGULARIZATION FOR MODEL-BASED REINFORCEMENT LEARNING

## ABSTRACT

In sequential tasks, planning-based agents have a number of advantages over model-free agents, including sample efficiency and interpretability. Recurrent action-conditional latent dynamics models trained from pixel-level observations have been shown to predict future observations conditioned on agent actions accurately enough for planning in some pixel-based control tasks. Typically, models of this type are trained to reconstruct sequences of ground-truth observations, given ground-truth actions. However, an action-conditional model can take input actions and states other than the ground truth, to generate predictions of unobserved *counterfactual* states. Because counterfactual state predictions are generated by differentiable networks, relationships among counterfactual states can be included in a training objective. We explore the possibilities of *counterfactual regularization* terms applicable during training of action-conditional sequence models. We evaluate their effect on pixel-level prediction accuracy and model-based agent performance, and we show that counterfactual regularization improves the performance of model-based agents in test-time environments that differ from training.

## 1 INTRODUCTION

Recent advancements in the use of variational inference and generative neural network architectures have made it possible to build accurate transition models for high-dimensional sequential decision-making environments. Networks have been trained to accurately predict future states conditioned on actions in large Markov decision process state spaces including Atari, VisDoom, and pixel-based robotic control tasks (Kurutach et al., 2018) (Ha & Schmidhuber, 2018) (Hafner et al., 2018). Using such a transition model, an agent can plan sequences of future actions to maximize an expected reward, using model-predictive control or other search algorithms. These approaches fall into the paradigm of *model-based* reinforcement learning (RL), distinct from the standard model-free approach of training a policy or value network. Model-based approaches have better sample efficiency than model-free methods, which usually require a large amount of training data to perform well.

There are distinct differences between traditional model-based RL, in which a transition between states are defined by a Markov transition matrix, and model-based approaches in which the state representation is learned, in which the state is represented as a continuous vector. This is made more unclear with environments that start with a scene described as an image, where it is well-known typical transition models that maximize a variational bound on the likelihood of future observations, similar to a variational autoencoder (Kingma & Welling, 2013) tend to generate blurry samples and exhibit posterior collapse (van den Oord et al., 2017), a phenomenon showing that the learned states may not be robust enough.

We hypothesize that a representation in which states are factored and well-separated will improve the performance of model-based agents. In order to achieve this, we propose a new family of regularizations applicable to action-conditional sequence prediction models, which we call *counterfactual regularization*. These regularizations are loss terms based on *counterfactual states*, which can be generated by feeding non-ground-truth actions or states into a transition model. Specifically, we propose regularization terms to achieve two goals. First, the set of possible states should be distinct in such a way that any change in input actions should result in a change in latent state. Second, that the dimensions of the learned state should be disentangled into separate factors in such a way that a change to one factor should effect only a sparse set of other factors.

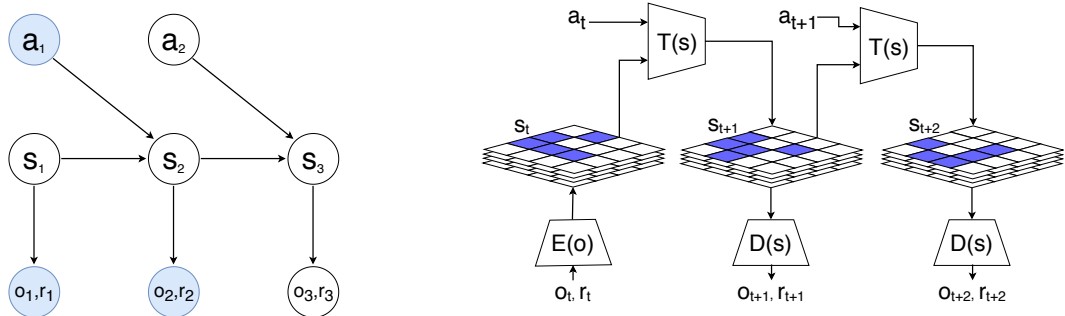

Figure 1: **Left**: The graphical model representing the generative process. Shaded nodes are observed, and empty nodes are not observed. In this example, the state at time $t = 2$ has been observed. **Right:** The latent state space $s$ is represented as a convolutional feature map, and the transition function $T(s)$ is implemented as a fully-convolutional neural network.

In order to test these regularizations, we introduce the *StarIntruders* game, a sequential environment with test-time dynamics that vary from training dynamics to test the generalization performance of trained agents. We propose a convolutional recurrent latent-state architecture well-suited for modeling two-dimensional video environments and trained using counterfactual regularization. Our experiments show that a simple planning agent using an action-conditional model trained with the proposed counterfactual regularization outperforms both non-regularized models, and a model-free agent, in test environments that differ from training.

## 2 LATENT DYNAMICS MODELS

### 2.1 PRELIMINARIES

We consider a partially-observable Markov decision process (POMDP) in which an agent observes at each time step $t$ an observation $o_t$ from an observation space $O$ and a reward $r_t$ from a reward space $R$ and executes an action $a_t$ from a discrete set of actions $A$. The agent executes actions from $t = 1$ until a terminal timestep $t = T$, generating a trajectory $(o_1, r_1, a_1)...(o_T, r_T, a_T)$. An optimal agent will take actions that maximize the cumulative reward $\sum_t^T r_t$ over the trajectory.

In pixel-based domains, $o_t$ may be a high-dimensional vector (e.g. $O = \mathbb{R}^{256x256x3}$) and the underlying dynamics function $p(o_t, r_t | o_{<t}, r_{<t}, a_{<t})$ may not be known. We are concerned with the problem of learning from experience a parameterized function $p_\theta$ that estimates the distribution of future observations and rewards conditioned on actions. We model the environment using a neural network similar to the deterministic state space model from (Hafner et al., 2018), trained on trajectories collected by an exploration policy.

Using such a function (a *model* of the environment), planning algorithms such as model-predictive control can be applied to compute an optimal action $a_t$ at each timestep. Unlike model-free reinforcement learning agents, a model-based agent does not learn a policy network or value function estimate; instead, actions are chosen by searching among possible future trajectories.

Previous work has explored the efficiency benefits of model-based methods, which can achieve similar performance to model-free methods using far fewer training examples (Hafner et al., 2018) (Kaiser et al., 2019b). Our work instead explores the capability of model-based agents to generalize to changes in the environment.

The graphical model in Fig. 1 illustrates our assumptions about the generative process. We model all dependencies on previous timesteps through an unobserved latent state variable $s_t$, which depends only on the previous state and action $s_{t-1}, a_{t-1}$. Each observation depends only on the current state. Accurately predicting future observations and rewards depends on building a highly accurate latent dynamics model.

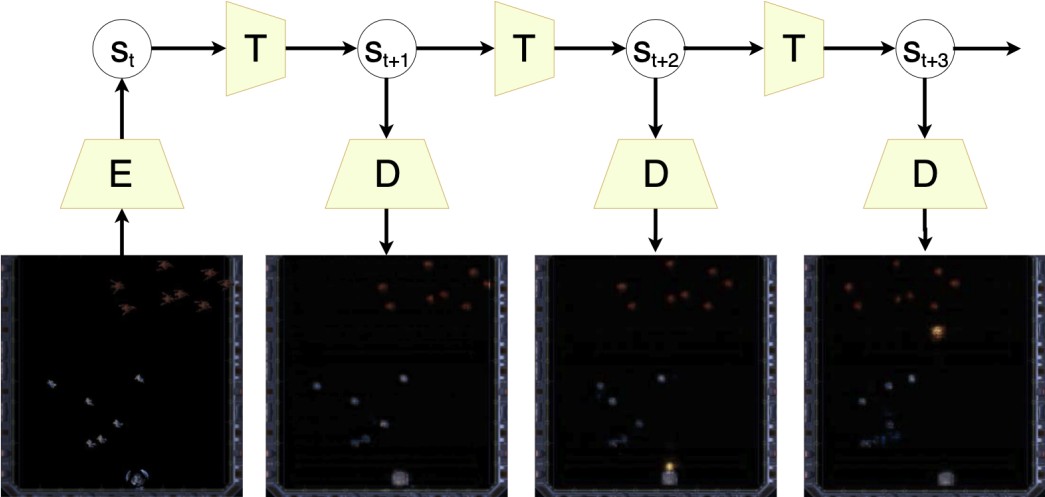

Figure 2: Example sequence predictions in the *StarIntruders* task. The encoder $E$ converts observed pixels to a latent state $s$. The recurrent transition function $T(s, a)$ predicts future latent states, and a decoder network $D$ decodes states back to pixel space.

## 2.2 MODEL ARCHITECTURE

Our latent dynamics model consists of the following components: an encoder $E : O \rightarrow S$, an action-conditional transition function $T : S \times A \rightarrow S$, a decoder $G : S \rightarrow O$, and a reward decoder $R : S \rightarrow \mathbb{R}^K$, as shown in Figure 1. Each component is a convolutional deep neural network with strided and transposed convolutions, and LeakyReLU nonlinearities. Spectral Normalization (Miyato et al., 2018) is applied to each convolutional layer of $T$ to assist in gradient propagation. For environments in which state estimation requires observation of multiple sequential states, the input layer of $E$ is resized to take as input a stack of $K$ sequential frames. Where not otherwise noted, $K = 3$.

The transition network $T$ converts each integer action input to a one-hot categorical representation, broadcast across the spatial input to the network. The reward network $R$ predicts a positive or negative reward at each timestep, for each spatial location, for each reward type. To improve training stability, reward estimates are discretized in the following way: each spatial location is classified as causing -1, 0, or +1 reward, and the output reward is the sum of all spatial locations.

## 2.3 TRAINING

The model is trained to reconstruct sequences of observations gathered from the true environment by an exploration policy. Given a sequence of ground truth observations, the model must reconstruct future observations and rewards (see Figure 2). Similar to (Hafner et al., 2018), our model consists of separate encoder/decoder networks which convert between observations and a learned latent representation, and a transition network which operates recurrently in the latent space (See Figure 1). This structure allows the latent representation to keep track of parts of the state not visible in current observations.

## 2.4 MODEL-PREDICTIVE CONTROL

Given a predictive model of future observations and rewards, we apply a simple model-predictive control algorithm to select actions at each time step during evaluation. The planning process produces a batch of $B = 64$ trajectories for each possible action $a^i$ with each trajectory taking a random rollout for a fixed horizon $T = 20$. The agent executes the action $a_t$ that maximizes the estimated sum of rewards $\sum_{i=t+1}^{t+T} r_i$, observes the next observation $o_{t+1}$ and then re-plans at each step.

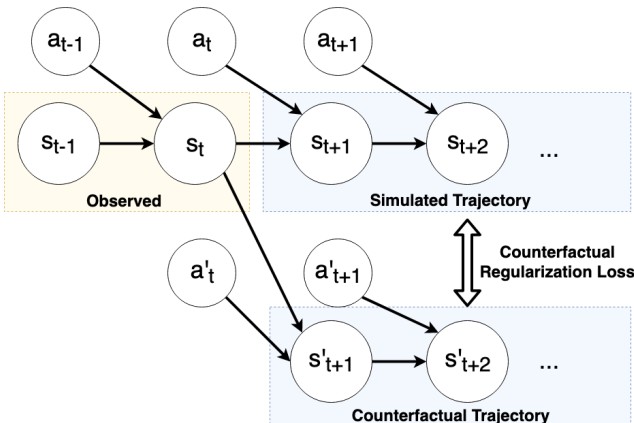

Figure 3: During training, in addition to predicting a ground truth sequence based on actions $a_t, a_{t+1}...$ our model predicts counterfactual trajectories based on alternate actions $a'_t, a'_{t+1}...$ for which no ground truth exists. By enforcing desired relationships between alternate simulated trajectories, desired causal relationships can be imbued into the model.

## 2.5 COUNTERFACTUAL TRAJECTORIES

A transition network $T$ takes as input a latent state $s_t$ and an action $a_t$, and produces an estimate of the successor state $\hat{s}_{t+1}$. The network is trained to simulate a ground truth trajectory $\{(o_1, a_1), (o_2, a_2)...(o_N, a_N)\}$ generated by a policy $\pi$ in the real environment.

The transition network is applied recurrently to generate a sequence of states:

$$s_{t+2} = T(s_{t+1}, a_{t+1}) = T(T(s_t, a_t), a_{t+1})$$

During training, $s_t$ is an observed state and $a_t$ is a known ground-truth action, selected by the training exploration policy. However, an alternate action $a'_t$ or an alternate state $s'_t$ can be input to the transition function, thereby generating an alternate or *counterfactual* state: $s'_{t+1} = T(s_t, a'_t)$ or $s'_{t+1} = T(s'_t, a_t)$. Repeated application of the transition function then generates a counterfactual trajectory:

$$s'_{t+2} = T(s'_{t+1}, a'_{t+1})$$

For many tasks, especially in physical environments, it may not be possible to repeat a previously observed trajectory, and so the "true" value of a counterfactual state may never be observable. However, the estimate $\hat{s}'_{t+1}$ is computable, and because $T$ is a differentiable parameterized function, the gradient of a loss function involving generated counterfactual states can be computed with respect to network parameters.

Relationships among estimates of real and counterfactual states provide a language by which statements about causality can be converted into differentiable loss terms suitable for inclusion in a neural network training process. Many useful desiderata can be posed in the language of counterfactuals, and converted to loss terms along with standard pixel-reconstruction and reward losses used in predictive models. By including a counterfactual loss term $\mathbf{L}_R$ in the objective function, the full loss becomes:

$$\mathbf{L} = \sum_t ||D(s_t) - o_t||_2^2 + ||R(s_t) - r_t||_2^2 + \lambda \mathbf{L}_R \tag{1}$$

where $D$ and $R$ are the decoder and reward prediction networks, $o_t, r_t$ is a ground truth observation and reward for timestep $t$, $\hat{s}_t$ is a predicted latent state output by one or more recurrent applications of $T$, $\lambda$ is a scalar hyperparameter, and $\mathbf{L}_R$ is one of the counterfactual regularization terms described below.

## 3 CAUSAL REGULARIZATION THROUGH COUNTERFACTUALS

By generating counterfactual trajectories during training (see Figure 3), we can augment the training objective with new terms. We explore a number of possible types of counterfactual regularization, given below.

### 3.1 ACTION-CONTROL REGULARIZATION

When exploring a complex environment with an untrained policy, it may be the case that the reward function can be accurately modeled as a deterministic function of time, unrelated to the agent's actions. For instance, in a timed game ending with a reward of -1 or +1 indicating win or loss, a poorly-initialized agent might learn that its actions have no effect on the environment, and may model the reward mechanism as a timer that outputs 0 or -1. This failure case could be considered analogous to the psychological phenomenon of *learned helplessness*, the perception of independence between actions and outcomes (Langer, 1975).

To address this problem, we might wish to limit our learning system to consider only models with the property that every action taken by the agent has a causal effect on the state. Using the language of counterfactuals, we can express this property precisely: for any state $s_t$ reached by taking actions $a_1, a_2, ...a_{t-1}$, there must exist some $a'_1, ...a'_{t-1}$ such that $s_t \neq s'_t$ and $\exists\, i < t,\ a_i \neq a'_i$.

Considering $i = t - 1$ for simplicity, we can convert the above constraint into a regularization term:

$$\min_{a'_{t-1}} -\log ||T(s_{t-1}, a_{t-1}) - T(s_{t-1}, a'_{t-1})||_1$$

where $s'_t = T(s_{t-1}, a'_{t-1})$ is a *counterfactual state* generated by the alternate $a'_i$. This regularization ensures that the learned environment model must include a causal link from $a_i$ to $s_t$, in the sense that for any $s_t$ there must exist a counterfactual action $a'_i$ which would have resulted in a different state $s'_t$.

In a more general case, the counterfactual $a'_i$ might be multiple time steps removed from $s_t$. Using $T(s, a_1...a_n)$ to denote recurrent application of $T$ to simulate multiple timesteps, the term can be expanded to the general case of $i < t - 1$.

$$\mathbf{L}_R = \min_{a'_i} -\log ||T(s_i, a_i...a_{t-1}) - T(s_i, a'_i...a'_{t-1})|| \tag{2}$$

For finite discrete action spaces, $a'_i$ can be minimized by exhaustive search, but for simple environments it suffices to generate $a'_i$ by sampling uniformly from $A \setminus \{a_t\}$. This loss term has the effect of biasing the transition model by forcing the latent state to change as a result of every agent action, which is an appropriate assumption for many control tasks.

### 3.2 DISENTANGLEMENT REGULARIZATION

Many practical environments contain separate and largely independent factors of variation. For example, in the game of *Space Invaders*, the agent controls one visible object on the screen (the player character) while enemy characters move across the screen, independently of the player. The only interaction between enemy position and player position is through projectiles fired by the player at enemies. It might be desired that a model of the game represent the position of enemies as independent of the position of the player, conditioned on projectiles.

In a non-sequential variational autoencoder setting, FactorVAE (Kim & Mnih, 2018) applies a regularization based on adversarial training using examples generated by permuting the dimensions of the latent representation of real examples. This forces the network to learn a representation such that any two latent dimensions could be swapped, and the result should still be realistic. The resulting representation maximizes the independence of latent dimensions.

Similarly but in a sequential environment, we desire a representation with the property that any two latent dimensions indices $k, j$ of a state $s_t$ can be swapped to create an alternate $s'_t$, and the resulting sequence of counterfactual states $s'_{t+1}, s'_{t+2}, ...$ would continue on with all dimensions that are independent of $k$ and $j$ unchanged. That is, changing the values of some dimensions of the latent representation should cause a sparse and minimal change in other latent dimensions.

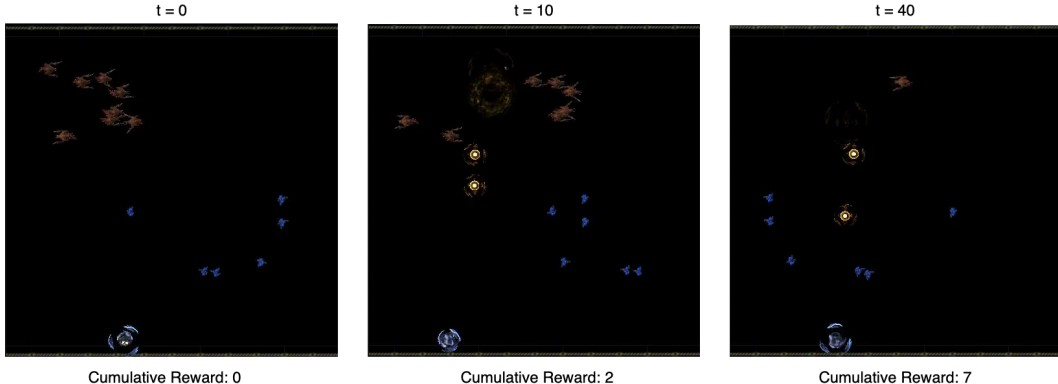

Figure 4: An agent completing the training version of the *StarIntruders* task. Destroying an enemy generates +1 reward, and destroying a human generates -1. A regularized model-based agent destroys fewer humans than a non-regularized model-based agent or a model-free agent, even in perturbed versions of the environment.

We represent this sparsity with an L1 regularization term:

$$\mathbf{L}_D = \sum_{i \notin \{j,k\}} |s_{t+H}^{(i)} - s_{t+H}'^{(i)}| \tag{3}$$

where $s^{(i)}$ denotes the $i$th dimension (or channel) of the latent state vector $s$. Here, $s_t = \left[ s_t^{(1)}, ... s_t^{(i)}, ... s_t^{(j)}, ... s_t^{(K)} \right]$ is the state vector at time $t$ and $s_t' = \left[ s_t^{(1)}, ... s_t^{(j)}, ... s_t^{(k)}, ... s_t^{(K)} \right]$ is the vector with dimensions $j$ and $k$ swapped. A fixed number of time steps $H$ is used between the counterfactual intervention and the regularization. The effect of this regularization is to promote sparse, rather than dense, relationships between latent factors. This is appropriate for environments where the underlying mechanics are discrete and deterministic.

## 4 EXPERIMENTS

### 4.1 ACTION-CONDITIONAL STATE PREDICTION

Our model is trained with ground-truth trajectories sampled by a random exploration policy. We train with $\lambda = .01$ and a maximum prediction horizon of $H = 10$ in both environments. After training, the model-predictive control (MPC) agent is applied to select the action with highest expected reward over the planning horizon at each time step.

### 4.2 GENERALIZATION EVALUATION ENVIRONMENT

We train and evaluate agents in the *Star Intruders* game, a custom sequential environment with mechanics similar to Space Invaders, built using the StarCraft II Learning Environment (Vinyals et al., 2017). The state space of the task includes a 256x256x3 RGB pixel matrix for visualization, and a 64x64x4 binary mask input to the network, indicating the positions of each type of object. The

|  | MSE H=3 | MSE H=5 | MSE H=10 | MSE H=20 | Avg. Game Score |
|---|---|---|---|---|---|
| **Ablation** | .0017 | .0021 | .0025 | .0027 | 4.43 |
| **Action-Control** | .0019 | .0023 | .0043 | .0072 | 5.11 |
| **Disentanglement** | .0015 | .0017 | .0020 | .0023 | 5.83 |

Table 1: MiniPacMan environment: Mean squared error (MSE) of predicted video frames at selected timestep horizons $H$. **Right Column**: Average score of the model-predictive control agent on 100 games. **Ablation** is trained with the standard reconstruction loss: each other network is trained with one of the proposed regularizations.

dynamics of the task are as follows (See Figure 4). The agent controls a unit at the bottom of the game map. Above the agent, randomly-positioned friendly and enemy units patrol left and right At each time step, the agent can take one of four actions: move left, move right, fire, or no-op. The *fire* action fires a projectile from the player's current position, which annihilates upon contact with any enemy or friendly unit.

When a projectile destroys an enemy unit, the agent receives a reward of 1, and when a projectile destroys a friendly unit, the agent receives a reward of -1. The optimal policy will fire projectiles aimed in such a way that all enemy units are destroyed but no friendly units are destroyed.

The training environment contains 8 friendly units and 8 enemy units. Each friendly unit begins at a random location in the bottom half of the map, and moves horizontally in a repeating pattern, from its starting location to the rightmost edge of the map, then to the leftmost edge, and so on. Each enemy unit patrols similarly from a random starting location in the top half of the map. Each episode continues until either all enemy units are destroyed, or a maximum number of time steps $T_M = 300$ are elapsed. The maximum cumulative reward in any episode is 8.

### 4.2.1 GENERALIZATION EVALUATION

After training, each policy is evaluated in three environments. The **Training** environment operates exactly as described in the previous section. The **Test A** environment operates in the same way as the training environment, except that instead of patrolling right-to-left, each unit will patrol either right-to-left or left-to-right with equal probability. The **Test B** environment operates in the same way as the training environment, except that instead of starting at random locations within the entire bottom half of the map, all friendly units are positioned within a small horizontal band close to the bottom of the map. From a subjective human perspective, the Test A and Test B environments are no more challenging than the training environment. Colors, textures, timing and input format in the test environments are identical to the training environment.

For comparison with model-free methods, we train a network with the Rainbow algorithm (Hessel et al., 2018) to maximize reward in the StarIntruders environment. The policy is trained for 1M time steps in the original training environment. The trained policy is evaluated in the original training environment, and in each variant version of the environment.

### 4.3 RESULTS

Table 2 shows game score in the StarIntruders training and test environments. Without regularization, the model-free Rainbow algorithm outperforms our model-predictive control agent in the training environment. However, the model-based method generalizes to new environments more effectively. Applying counterfactual regularization to the predictive model improves the performance of our MPC agent in both the training environment and test environments.

Table 1 shows both game score and pixel prediction error for the simpler MiniPacMan task. We note that lower MSE does not necessarily coincide with higher score, so the proposed regularizations do not simply improve pixel prediction accuracy. We conclude that counterfactual regularization improves the representation learned by our environment model.

| Method | Training | Test A | Test B |
|:---:|:---:|:---:|:---:|
| **Rainbow** (Hessel et al., 2018) | 7.38 | 4.92 | 5.05 |
| **MPC** | 6.50 | 7.30 | 7.00 |
| **MPC**, Action-Control | 7.60 | 7.70 | 7.60 |
| **MPC**, Disentanglement | 7.90 | 7.90 | 8.00 |

Table 2: StarIntruders environment: Per-episode cumulative reward for training and test tasks, average of 10 episodes. The same model-predictive control (**MPC**) agent is applied using models trained with the standard reconstruction loss, and models trained with each proposed regularization. A model-free method, **Rainbow** is included for comparison.

## 5 RELATED WORK

Much previous work in model-based reinforcement learning involves planning with model-predictive control in low-dimensional state spaces using neural networks (Gal et al.) (Chua et al., 2018) or other models (Deisenroth & Rasmussen, 2011). Other work has applied generative neural network models to high-dimensional video prediction (Chiappa et al., 2017) or state prediction in Markov reward processes (Silver et al., 2017) where transitions are not action-conditional. Some work has explored action-conditional video prediction (Oh et al., 2015) without planning. Recent work has applied model-predictive planning to high-dimensional image domains (Kaiser et al., 2019a) (Hafner et al., 2018).

### LEARNING WITH AUXILIARY REWARDS

Related to model regularization, some work applies auxiliary tasks during training of a model-free or model-based agent with the goal of improving performance or generalization. In (Jaderberg et al., 2016) a representation is learned using auxiliary policies based on intrinsic rewards such as maximizing changes in pixel intensity, in a model-free setting. In a model-based setting, in (Pathak et al., 2017) a representation of the environment state is trained using inverse or forward dynamics, and the prediction error of this dynamics model is subsequently used as an intrinsic reward signal to train a model-free policy network.

### DISENTANGLEMENT OF FACTORS

Factorized state representations have long been known to improve performance in reinforcement learning tasks (Degris et al., 2006). In (Higgins et al., 2017), the Kullback-Liebler divergence term of a variational autoencoder's objective function is reinterpreted as a regularization term promoting conditional independence of latent dimensions. In (Kim & Mnih, 2018), independence of an autoencoder's latent dimensions is achieved via an adversarial loss.

### MODEL-BASED REINFORCEMENT LEARNING

Learning a predictive model of the environment has long been recognized as a more sample-efficient approach than model-free policy learning. Predictive models were applied to discrete grid-world tasks in Dyna-Q (Sutton, 1990), (Chentanez et al., 2005) and to low-dimensional robotic control tasks in PILCO (Deisenroth & Rasmussen, 2011). In (de Avila Belbute-Peres et al., 2018), a linear complementary problem solver is used to simulate two-dimensional rigid body dynamics as a differentiable layer within a network, allowing sample-efficient prediction and high performance on control tasks. Other methods build models of environment dynamics which are then used to plan for control tasks (Lowrey et al., 2018) (Chua et al., 2018). Many recent methods use generative deep network models trained to predict transitions in high-dimensional visual environments. Planning-based agents have been constructed using networks trained to simulate grid-based games such as Sokoban (Racanière et al., 2017), Atari games (Kaiser et al., 2019a), pixel-based physics tasks (Hafner et al., 2018) and visual robotic control tasks (Ebert et al., 2018).

## 6 CONCLUSIONS

The capability to generate and reason about counterfactual states is already considered valuable when deploying agents that require interpretability or explainability (Wachter et al., 2017). We have shown how counterfactual states can also be used during the training process, in order to force desired causal dependencies or independencies that may be appropriate for a given task or environment.

We have proposed counterfactual regularization terms appropriate for two-dimensional video tasks, but loss functions based on counterfactual states provide a flexible and general way to encode desired properties into a loss function. Future work might explore new applications of counterfactual states to enforce desired causal relationships during the training process. Additional future work includes extending counterfactual regularization to new architectures and more complex physical control tasks, as well as considering more sophisticated exploration policies for generating training data, and better planning algorithms at test time.

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
