# OpenReview forum: "Counterfactual Regularization for Model-Based Reinforcement Learning"
_ICLR.cc/2020/Conference — Reject_

### Official Review · AnonReviewer2 · 2019-10-21
**Official Blind Review #2**

**Rating:** 3

**Review:**

The paper presents regularization techniques for model based reinforcement learning which attempt to build counterfactual reasoning into the model. In particular, they present auxiliary loss terms which can be used in "what if" scenarios where the actual state is unknown. Given certain assumptions, they show that this added regularization can improve generalization to unseen problem settings. Specifically they propose two forms of regularization: (1) enforcing that for different actions the predicted next state should be different (action-control) and (2) enforcing that when certain parts of the low dimensional state are perturbed, over a model rollout the perturbation should only affect the perturbed parts of the state, essentially encouraging the latent space features to be independent (disentanglement).

Overall the idea is well motivated - incorporating counterfactual reasoning into model based RL has potential to to improve generalization. Also, while the assumptions needed for the regularization to be correct are not always true, they do seem to hold in many cases. Lastly, the results do seem to indicate that generalization is slightly improved when using the proposed forms of regularization.

My criticisms are:

(1) As mentioned in the paper Action-Control assumes that at every single timestep the agent has potential to change the state. However there may be settings where the agent can always change state, but only a small component of the state. In these cases the states should be quite similar. For example a robot only moving a single object when the state consists of many objects. Also as mentioned in the paper Disentanglement will not work in stochastic environments. One concern I have is that since different environments can violate the assumptions to varying degrees, it seems like actually using the regularization and picking the correct hyperparameter to weight it will be very challenging.

(2) The current results are only demonstrated in a single, custom environment. Additionally performance is shown on only 2 test tasks, and in all cases in Table 2 it is unclear how to interpret the reward. Does this performance constitute completing the task? What is the best possible cumulative reward in this case? The performance improvement seems small, but it is difficult to judge without knowing the details of the task.

I think the paper would be significantly improved by (1) adding experiments in more environments, especially standard model based RL environments where the performance of many existing methods is known and (2) adding comparisons to other forms of model regularization, for example using an ensemble of models. My current rating is Weak Accept.

Some other questions:
- In Table 2 does MPC amount to PlaNet?
- How sensitive are the current numbers to planning parameters (horizon, num samples)?
- Can you provide error bars for the numbers in the tables?

______________________________________________

After author responses and closer examination of the paper I have some additional concerns about experimental details.  Changing my score from 'Weak Accept' to 'Weak Reject'

**Experience Assessment:**

I have published one or two papers in this area.

**Review Assessment: Checking Correctness Of Derivations And Theory:**

N/A

**Review Assessment: Checking Correctness Of Experiments:**

I assessed the sensibility of the experiments.

**Review Assessment: Thoroughness In Paper Reading:**

I read the paper at least twice and used my best judgement in assessing the paper.

---

### Official Review · AnonReviewer3 · 2019-10-23
**Official Blind Review #3**

**Rating:** 3

**Review:**

This paper considers regularization based on "counterfactual" trajectories.
Namely, it suggests two losses, action-control and disentanglement regularization.
It experimentally evaluates the benefits of such regularization in the StarIntruders environment.

The paper is well written and explained.

Issues:

1) Authors evaluated the two suggested regularizations in separate.
I would like to also see numbers from a combination of these.

2) I think the related work is missing a large line of work on "auxiliary tasks".
It seems to me that this paper would exactly fit within that scope?

3) My main issue is the evaluation.
The evaluation is done on a in-house game and compares to very few methods.
For a paper that has very little theory and thus most of the value is in the empirical evaluation, I think that is a problem.
If authors opted for example for Space Invaders (they do say it is similar) or simply more games, one would have many more existing numbers to compare against.

Minor issues:

1) The first regularization - action control regularization is motivated by the idea that there is always an action that changes the state. While true for most environments, this does not hold in general.

Summary:

Overall, this paper has potential but I don not believe is good enough - I suggest a reject.
The main problem is that the idea is relatively simple, there is no theory and thus the crucial piece of the paper has to be the empirical evaluation.
And the evaluation only compares to a single method with no regularization, no auxiliary tasks and reports only experiments on a single game.

**Experience Assessment:**

I have read many papers in this area.

**Review Assessment: Checking Correctness Of Derivations And Theory:**

I assessed the sensibility of the derivations and theory.

**Review Assessment: Checking Correctness Of Experiments:**

I carefully checked the experiments.

**Review Assessment: Thoroughness In Paper Reading:**

I read the paper at least twice and used my best judgement in assessing the paper.

---

### Official Review · AnonReviewer1 · 2019-10-25
**Official Blind Review #1**

**Rating:** 3

**Review:**

This paper proposes a method called "counterfactual regularization" whereby the dynamics/transition model is encourage to not have degeneracies where the actions don't influence the state transitions.  Concretely, this is done by, for every state, computing the maximum deviation of Transition model under a different action than the action taken in the history, and encourage that deviation to be as large as possible.  If that maximum deviation is 0, then all actions lead to the same next state. Empirical results show reasonable improvements in the StarIntruders task.

My biggest complaint (and the only one barring me from supporting acceptance) is that I don't see the body of results as scientifically solid.  Example of additional results that I would find much more convincing are:

-- Experiments on more than one environment.  Currently, this paper should be judged solely for its empirical improvements, because there is little formal analysis or rigorous derivations.  But it's hard to judge that based on only one experiment.

-- Deeper investigation into the effects of counterfactual regularization, including its interaction with learning disentangled representations.  Right now, there is no investigation, just a numerical score of reward attained.  This does not lead to much scientific insight.

-- Exploration of the limitations of the approach.  Personally, I think this approach is reasonable for video games with a few discrete actions, but quickly runs into problems for more complex action spaces.

**Experience Assessment:**

I have read many papers in this area.

**Review Assessment: Checking Correctness Of Derivations And Theory:**

I carefully checked the derivations and theory.

**Review Assessment: Checking Correctness Of Experiments:**

I carefully checked the experiments.

**Review Assessment: Thoroughness In Paper Reading:**

I read the paper thoroughly.

---

### Author Response · Authors · 2019-11-15
**General Response**


We would like to thank the reviewers for taking the time to review the paper and for their insightful feedback.

Regarding environment dependency, we agree with the view that our proposed regularizations are environment-dependent; state and action spaces as well as an agent's ability to control the environment may vary.
We do find some sensitivity to training hyperparameters, as well as planning horizon, and consistent with the reviewers' intuitions we would expect some level of task-specific hyperparameter tuning to be required to achieve optimal results for most tasks.

Regarding the case of action-control regularization in an environment where an agent's action may not always have an impact on the observed state, we would note that the regularization enforces a difference between learned latent states, not necessarily between observations.
It is possible in principle for a transition model to learn to produce different latent states given different agent actions, even if the latent states produce the same observations.

Regarding the relationship between the planning algorithm used in our experiments and PlaNet, MPC in table 2 is not exactly equivalent to PlaNet. Although our model is similar in structure to Hafner et al.[1], their planning approach is based on the Cross Entropy Method [2] while we use a simpler deterministic search (select the action leading to maximum predicted reward over a fixed horizon).

Regarding the details of the generalization test environment, the maximum score in every version of the task is 8.0, and this has been clarified in section 4.

Regarding some of the related work mentioned by reviewer #3, we agree on the relevance of the literature on auxiliary tasks and have updated the related work section accordingly.

[1] Hafner, Danijar, et al. "Learning latent dynamics for planning from pixels." arXiv preprint arXiv:1811.04551 (2018).
[2] Chua, Kurtland, et al. "Deep reinforcement learning in a handful of trials using probabilistic dynamics models." Advances in Neural Information Processing Systems. 2018.

---

### Decision · Program_Chairs · 2019-12-19

**Decision:**

Reject

**Comment:**

I agree with the reviewers that this paper has serious limitations in the experimental evaluation.